# Characterization of HRRR simulated Rotor Layer Wind Speeds and Clouds along Coast of California

Jungmin Lee<sup>1</sup>, Virendra P. Ghate<sup>2</sup>, Arka Mitra<sup>2</sup>, Lee M. Miller<sup>3</sup>, Raghavendra Krishnamurthy<sup>3</sup>, and Ulrike Egerer<sup>4</sup>

<sup>1</sup>Lawrence Livermore National Laboratory, Livermore, CA 94550, USA

Correspondence to: Jungmin Lee (lee1046@llnl.com)

Abstract. Stratocumulus clouds, with their low cloud base and top, affects the atmospheric boundary layer wind and turbulence profile, thereby modulating wind energy resources. GOES satellite data reveal an abundance of stratocumulus clouds in late spring and summer months off the coast of Northern and Central California, where there are active plans to deploy floating offshore wind farms at two lease areas (near Morro Bay and Humboldt). From fall 2020, two buoys equipped with multiple instruments including Doppler lidar were deployed for about one year in these wind farm lease areas to assess the rotor-layer wind conditions in these locations. The objective of this study is to evaluate how well the High Resolution Rapid Refresh (HRRR) model represents stratocumulus cloud characteristics and turbine-relevant rotor-layer winds (surface to 300 m) by comparing HRRR simulations with buoy and satellite observations. We first find that the HRRR model reproduces the seasonal cycle of cloud-top height reasonably well in these regions. However, during the warm season—especially at Morro Bay—the HRRR-simulated stratocumulus clouds tend to have lower tops by about 150 m and exhibit weaker diurnal cycles than satellite observations. Our analysis also shows that rotor-layer wind speeds and vertical shear are stronger at Humboldt than at Morro Bay, and both are generally stronger under clear-sky conditions. Finally, the HRRR model bias in rotor-layer wind speed is small under cloudy conditions but larger and dependent on observed wind speed under clear skies. Specifically, HRRR underestimates wind speeds at Morro Bay and overestimates them at Humboldt under clear-sky conditions.

## 1 Introduction

25

Due to the semi-permanent Pacific high-pressure system and cold sea surface temperatures (SST) from coastal upwelling, marine boundary layer clouds appear frequently off the coast of California year-round (Iacobellis et al., 2013, Lin et al., 2009). Especially during the warm season, marine stratocumulus and stratus (from here on, stratocumulus) are most pronounced. These clouds originate in subsiding dry and warm air as the high-pressure system interacts with the cold, humid, and shallow marine atmospheric boundary layer (MABL), forming a stronger capping inversion. This capping temperature and moisture inversion limits the vertical extent of the low-level clouds. Marine stratocumulus clouds are shallow (~200 m

<sup>&</sup>lt;sup>2</sup>Argonne National Laboratory, Lemont, IL 60439, USA

<sup>&</sup>lt;sup>3</sup>Pacific Northwest National Laboratory, Richland, WA 99354, USA

<sup>&</sup>lt;sup>4</sup>National Renewable Energy Laboratory, Golden, CO 80401, USA

thickness) but optically thick, therefore inducing radiative cooling at the top of the boundary layer, primarily in the longwave spectrum. The cloud top radiative cooling along with surface buoyancy production, cloud top entrainment, and wind shear drive the turbulence in the MABL and hence the cloud and precipitation processes. As turbulence and cloud processes are intimately coupled in the MABL, the presence of marine stratocumulus affects the profile of winds and turbulent properties within the marine boundary layer (Wood 2012).

Recent modeling and observational studies have provided deeper insight into the physical processes governing the stratocumulus-topped boundary layer (STBL). In particular, Kopec et al. (2016) investigated how radiative cooling at the cloud top and wind shear across the capping inversion jointly influence turbulence generation and cloud-top structure using large-eddy simulations based on the POST field campaign. Their analysis demonstrated that radiative cooling intensifies convective circulations within the STBL, while wind shear enhances turbulence and mixing in the inversion layer above the cloud, often producing a distinct turbulent sublayer that is dynamically decoupled from the convective motions below. These findings highlight the importance of representing both shear- and radiation-driven turbulence for accurate modeling of stratocumulus dynamics and their influence on boundary-layer winds. Such processes are particularly relevant for offshore wind resource assessment, where low-level clouds modulate turbulence and wind shear within the turbine rotor layer.

In July 2005, Marine Stratus/Stratocumulus Experiment (MASE) field campaign was carried out off the coast of Monterey (California) to sample the clouds and aerosols (Lu et al., 2007). Measurements including cloud base and top heights were made from research flights 13 different times. Out of those measurements, cloud base height varied between 67 and 315 meters, and the cloud top height ranged from 268 to 732 meters. These measurements indicate that all or a lower portion of marine boundary layer clouds can exist in the wind turbine relevant height near the coast of California in summertime. Therefore, it is important to evaluate the influence of marine boundary layer clouds on the wind resources off the coast of California (Shaw et al., 2022).

45

60

Off the coast of California, there are active plans to deploy offshore wind farms at two lease areas, one along northern California near Humboldt Bay and another along central California near Morro Bay. Starting in October 2020, Doppler lidar-equipped buoys that are owned by the US Department of Energy (DOE) were deployed to continuously measure the wind speed at various heights below 240-meter for one year (Krishnamurthy, 2023). This dataset provides a unique opportunity to evaluate the performance of numerical weather prediction (NWP) models at the planned wind farm locations. Sheridan et al. (2022) utilized this dataset to assess the bias in hub-height wind speed using various reanalysis and NWP model data. They also investigated potential relations between the bias in terms of atmospheric stability and shear strength. However, they did not consider the influence of MABL clouds as a source of model bias of the hub-height wind speed. Additionally, their analysis did not include an evaluation of winds simulated by the High-Resolution Rapid Refresh (HRRR) model, despite HRRR being one of the most widely utilized forecasting tools for evaluating skill in representing rotor-layer wind conditions over the United States.

Several prior studies have evaluated the HRRR model relevant to hub-height wind speed assessment. Liu et al. (2025) benchmarked hub-height wind speeds from HRRR analyses against multi-source observations across the southeastern United

States and found that wind-speed biases were strongly influenced by local topography and land surface characteristics such as forest canopy height. Complementary efforts under the Second Wind Forecast Improvement Project (WFIP2) assessed experimental updates to the HRRR model using scanning Doppler lidar measurements in the complex terrain of the Columbia River Basin (Pichugina et al., 2020). That study demonstrated that the HRRR model's wind-speed errors were largest below about 150 m above ground level and that improvements in model physics and grid resolution led to modest but consistent reductions in mean wind-speed bias. Together, these findings highlight the importance of evaluating HRRR performance in diverse environments, including offshore regions where boundary-layer cloud processes can further influence rotor-layer winds.

The aim of this study is to compare the modeled (HRRR) and observed (satellite and buoy) characteristics of stratocumulus clouds and winds at turbine relevant height and assess the HRRR bias in wind speed near the surface for better decision making in wind farm development off the coast of California. The paper is organized as follows: section 2 describes the brief description about the various datasets used in this study and how they were post-processed to allow direct comparisons. The characteristics of clouds, hub-height wind speed, and model wind bias in relation to cloud properties at Humboldt and Morro Bay locations are explored in Section 3. Summary and conclusion follow in Section 4.

# 2 Data and Method

#### 2.1 Lidar-buoy observation dataset

DOE-owned buoys with multiple instrumentations were deployed about 40-50 km off the coast of Morro Bay (35.71074° N, 121.84606° W) and Humboldt (40.9708° N, 124.5901° W). Each of the buoys started the operation in Fall of 2020 and the mission lasted about 1 year. In December 2020, a significant wave event at the Humboldt buoy location caused a power outage, resulting in a substantial data gap. The buoy equipment was restored and brought back online on May 25, 2021. The details about the instruments on the buoys and methods for the raw data processing can be found in Krishnamurthy et al. (2023).

The buoys were equipped with WindCube 866 Doppler lidar to measure the vertical profiles of motion-compensated wind speed at 1 second temporal resolutions. The final post-processed wind profiles are at 10-minute temporal resolution with 10 m range resolution from 40 to 240 meter above the surface.

The buoys were also equipped with LI-200SA pyranometer (PYR) to measure the global broadband solar radiation at 10-minute temporal frequency. By comparing the measured solar radiation to the modeled solar radiation for clear-sky conditions, we determined whether cloud presence attenuated the downwelling solar radiation reaching the surface (Long and Ackermann, 2000). The temporal cloud mask, which provides a timeseries of sky conditions, was derived by quantifying the deviation between the observed and modelled clear-sky collar radiation. When the measured radiation was more than 10% lower than the modeled clear-sky value, the condition was classified as cloudy. Otherwise, it was designated as cloud-free (Krishnamurthy et al., 2023).

## 2.2 Cloud top height from GOES







Cloud top heights over our study duration was taken from the SatCORPS Ed4 CERES-GOES record (Minnis et al. 2021). This record of ~8-km spatial resolution, hourly cloud property retrievals are prepared by the NASA Satellite ClOud and Radiation Property Retrieval System (SatCORPS) through the processing of GOES-15, GOES-16, and GOES-17 geostationary imagers belonging to the Geostationary Operational Environmental Satellite (GOES) program (Menzel and Purdom, 1994). For this study, GOES-17 is the only satellite providing data that overlaps with the buoy observation period. These retrievals use the Clouds and the Earth's Radiant Energy System (CERES) Edition 4 (Ed4) algorithms, adapted for geostationary applications. These Ed4 algorithms and their nominal quality assessments can be found in Minnis et al. (2021). Since these retrievals from geostationary satellites are not provided over a fixed spatial grid, we prepared a time series of hourly retrievals that were closest to the buoy locations (Section 2.1). From here on, we shall simply refer to this record as GOES.

GOES determines cloud-top height (CTH) from infrared-retrieved cloud-top temperature (CTT) and lapse-rate calculations following Sun-Mack et al. (2014). The CTT measurements have an accuracy of about 1 K (Yu et al., 2012), which corresponds to an uncertainty of roughly 100 m in CTH for a dry-adiabatic lapse rate. Previous studies have shown that inaccuracies in lapse-rate estimates can lead to CTH errors of this magnitude (Zuidema et al., 2009; Ghate et al., 2019). Comparisons of the most recent SATCORPS Edition 4 GOES retrievals with CloudSat and CALIOP indicate mean differences near 100 m (Yost et al., 2021). A recent seven-year analysis comparing ERA5 and GOES data reported ERA5–GOES differences of  $-0.17 \pm 0.62$  km (Humboldt) and  $-0.22 \pm 0.51$  km (Morro Bay) (Mitra et al., 2025). These results imply that GOES retrieval uncertainties are on the order of 100–200 m—smaller than the HRRR–GOES differences reported here—and that the HRRR bias in CTH likely reflects model rather than retrieval limitations.

Instances of multilayer cloud systems (e.g., thin cirrus overlying low stratocumulus) were handled by filtering out retrievals with cloud-top heights exceeding our low-cloud thresholds. In such cases, the GOES infrared retrieval reports the upper-level cirrus as the primary cloud top, leading to spuriously high CTH values that are automatically excluded from the analysis. Situations with optically thick cirrus that fully obscure the low-level cloud deck are likewise classified as high-cloud conditions and are not used when evaluating boundary-layer properties or HRRR wind-speed biases.

## 2.3 High Resolution Rapid Refresh (HRRR) v4

HRRR is a 3-km spatial resolution, hourly updating, convection permitting atmospheric forecast model developed and operated by National Oceanic and Atmospheric Administration (NOAA) (Dowell et al., 2022). HRRR provides high fidelity short-term atmospheric forecasting and provides the meteorological variables necessary for wind power estimation and forecasting applications. HRRR's initial conditions are generated from the parent Rapid Refresh (RAP) model one hour prior to forecast initialization. RAP forecast is advanced forward, assimilating the most recent observations to provide a dynamically consistent state for HRRR initialization. Lateral boundary conditions for HRRR are supplied by RAP model at three-hour intervals. HRRR employs a hybrid ensemble-variational (EnVar) data-assimilation system that integrates a 36-member

ensemble to represent flow-dependent background errors. The assimilation cycle updates every hour and incorporates a wide range of conventional, satellite, and radar observation. Within each cycle, radar reflectivity and radial-velocity data are assimilated every 15 minutes to improve the representation of cloud and convective structures. Surface and soil states are updated using short-term forecasts from the HRRR Data Assimilation System (HRRRDAS), which maintains temporal continuity between analysis cycles. This framework is designed to minimize spin-up errors and improve the depiction of boundary-layer and mesoscale processes critical to wind-forecast applications. We use data from HRRRv4, which has been released in December 2020 and its data availability aligns with our buoy observation period. We utilize HRRR 3-hour forecast data that outputs wind speed at the native model vertical grid. By comparing the model's vertical grid with the lidar buoy's measurement height, we determine that the first 4 of the model's vertical height levels are close to the heights of 40, 80, 160, and 240 meters at which data from lidar measurement is available. The HRRR reported wind profiles were linearly interpolated to get the wind speeds at 40, 80, 160, and 240 meters. We extract the data from the closest point to the buoy locations, which is not farther away than 1.4 – 2.5 km on average. We compute the wind speed bias by subtracting buoy reported wind speed from those reported by HRRR at 40, 80, 160, and 240 meter. Because HRRR outputs are available only at full-hour intervals, we selected the 10-minute averaged lidar measurements whose timestamps correspond exactly to each forecast hour, ensuring consistent temporal alignment between the HRRR time series and the buoy observations.

#### 3. Results






### 145 3.1 Characteristics of clouds off the coast of California

Figure 1(a,b) presents the distribution of cloud top heights computed by HRRR and estimated by GOES for each month of the observation period, and Figure 1 (c, d) show the same for cloud fraction. There are significant seasonal changes in cloud top heights and cloud fraction at both the Morro Bay and Humboldt locations. For Morro Bay, cloud fraction values from both HRRR and CERES exceed 80% consistently between May and September. In contrast, Humboldt Bay exhibits greater month-to-month variability in cloud fraction, with summer months not necessarily showing high values. The low cloud top heights (< 3 km) and high cloudiness between April and October, suggest the clouds to be low-level marine stratocumulus clouds. While higher cloud top heights (>7 km) and low cloudiness during the colder months suggest passage of mid-latitude frontal systems consistent with Lin et al., 2009. Overall, the winter months have higher cloud top heights than the summer months at both locations. This seasonal variation in cloud top height is accurately captured by the HRRR model. However, there are some minor discrepancies between the median values of observed and HRRR simulated cloud top heights. The quartile ranges and median values of cloud fraction reveal discrepancies between CERES and HRRR, with the differences being particularly pronounced for Humboldt Bay. These differences may partly reflect the distinct definitions of cloud used in the two datasets. The GOES retrieval identifies a cloud layer only when the optical depth exceeds a threshold that varies with viewing geometry and atmospheric conditions, while HRRR defines clouds based on the simulated presence of hydrometeors

within model grid cells. Because no satellite instrument simulator was applied to the HRRR output, such definition and sensitivity differences might contribute to the observed discrepancies in cloud-top height and cloud fraction.


As we are interested in investigating the HRRR bias when low-altitude clouds interfere with the rotor layer, rest of the analysis only uses data from the warmer months. Guided by Fig. 1, the data from May through the end of September at both Morro Bay and Humboldt is composited. Due to the power outage at Humboldt location, our results at the Humboldt location are missing most of May data.

Figure 1 Box plot of (a, b) cloud top height from GOES (red) and HRRR (blue) and (c, d) cloud fraction from CERES (red) and HRRR (blue) for each month of the observation period at (left column) Morro Bay and (right column) Humboldt. The box represents 25<sup>th</sup>, and 75<sup>th</sup> percentile, and the bounding lines represent the minimum and maximum values. Median values are represented by thick lines.


Figure 2 shows the diurnal variation of MABL cloud top heights (cloud tops below 1.5 km) from HRRR and GOES over the warm season (May-Sept). On average, HRRR cloud tops are about 150 m lower than those in the GOES record at Morro Bay, while there is little difference at Humboldt location. The mean cloud top estimated by GOES is about 700 m at

Morro Bay and 520 m at Humboldt, whereas HRRR estimates a mean cloud top height of around 550 m for both locations. A clear diurnal cycle in GOES cloud top height is present at either locations, which peaks around 10 AM local time (UTC - 8hr). The diurnal amplitude is larger for Morro Bay, while HRRR fails to reproduce the observed diurnal cycle at either site. Under typical marine stratocumulus conditions, cloud tops are higher at night and lower during the day due to stronger nocturnal cloud-top radiative cooling, our GOES analysis (Fig. 2) shows the opposite behavior at both sites; the reason for this discrepancy is presently unclear. The absence of a diurnal cycle in cloud top height in the HRRR model nonetheless suggests an inaccurate representation of the diurnal cycle of cloud top radiative cooling-driven turbulence in the HRRR model.

One likely contributing factor is the model's underrepresentation of cloud liquid water path (LWP), which cannot be directly compared with GOES in this study. A low simulated LWP would reduce the amount of shortwave absorption within the cloud during the daytime, resulting in insufficient diabatic heating and limited cloud deepening. Conversely, too little condensate would also weaken longwave cooling at night, reducing turbulent mixing at cloud top. These combined effects could explain the muted diurnal variation in HRRR cloud-top height and indicate that biases in the model's treatment of cloud microphysics and radiative processes jointly contribute to the discrepancy.


Figure 2 Warm season (May-Sept) diurnal cycle of cloud top height as reported by HRRR (gray) and GOES (blue) at (a) Morro Bay and (b) Humboldt Bay. Bounding boxes represent 25<sup>th</sup> and 75<sup>th</sup> percentile, while the lines represent the mean values.

Figure S1 shows the percentage of clear sky and cloudy sky days estimated by buoy, GOES, and HRRR between May-September at Morro Bay and Humboldt. For the buoy data, we utilize the estimated cloud mask, whereas for HRRR and GOES datasets, the cloud mask is derived from the percentage fraction of clouds, calculated as the proportion of all hours with valid cloud-top information. Out of 8,547 (11,951) buoy data samples collected at Morro Bay (Humboldt) during the warm season, 2,225 (4,529) were classified as clear-sky, while 6,322 (7,422) were classified as cloudy. Figure S1 indicates during warm season when the marine stratocumulus clouds are most pronounced, cloudy days are dominant off the coast of California and Morro Bay is cloudier than Humboldt, which is consistent with the finding in Krishnamurthy et al. (2023). HRRR-estimated cloud fraction slightly underestimates the buoy observed cloud fraction, which is less than 10 %, but is generally in good agreement with other sources of observation (GOES and buoy).

Throughout this study, we will investigate the HRRR bias in rotor layer wind speed in terms of cloudy days versus clear-sky days. Unless specified otherwise, only the warm season (May-Sept) data is considered in this study. Additionally, we are selecting a few case studies of marine stratocumulus clouds later in section 3.4 to investigate the model performance.

# 3.2 Variation of rotor layer wind speeds with cloudiness

Figure 3 PDF of rotor layer (<= 240m) wind speed at Morro Bay from (top) HRRR, (bottom) lidar buoy during (left) all conditions, (middle column) cloudy conditions, and (right column) clear sky conditions.

In Fig. 3, we present the probability density function (PDF) distribution of wind speed at Morro Bay at 40, 80, 160, and 240 m heights from HRRR (a-c), and from the lidar buoy observation (d-f). We used the observed cloud mask from the buoys to distinguish between clear-sky and cloudy conditions. Rotor layer wind speeds from the lidar buoy at Morro Bay peaks around 1 - 2 m s<sup>-1</sup> during cloudy condition, which is below typical cut-in speeds (3 – 4 m s<sup>-1</sup>), while it peaks around 10 - 15 m s<sup>-1</sup> during clear-sky conditions. Additionally, the peak of the PDF shifts towards higher wind speeds for clear-sky conditions and not for the cloudy conditions, suggesting presence of wind shear during the clear-sky conditions that is absent during cloudy conditions. This suggest that the turbulence is forced by wind shear during cloud-free conditions, while turbulence is only forced by cloud processes during cloudy conditions due to absence of shear. The rotor layer wind speed distributions at Morro Bay are well reproduced by HRRR model simulations with minor differences. During cloudy conditions, HRRR model does show a small increase in the wind speeds with height during cloudy conditions at higher wind speeds unlike the observations. During clear-sky conditions, the HRRR model did not simulate winds stronger than 25 m s<sup>-1</sup>, that were reported on few instances by the lidar.

Figure 4 Same as Fig 3, but for Humboldt location.

In Fig. 4, we repeated the analysis shown in Fig. 3 for Humboldt Bay location. At Humboldt, the lidar buoy measurements of wind speed in the rotor layer shows a more spread-out distribution spanning 0 – 20 m s<sup>-1</sup>, while HRRR reported winds range from 0 to 30 m s<sup>-1</sup>. The HRRR model simulates a peak in the PDF (Figure 4a) that shifts toward higher speeds with increasing heights, unlike what is observed (Figure 4d). At Humboldt, strong vertical shear is present regardless of cloud conditions, as the increase in wind speed with height is evident in both datasets, but HRRR model consistently overestimates this increase. During clear-sky conditions, the wind speed tends to be stronger compared to cloudy conditions, with peaks around 10 – 18 m s<sup>-1</sup>. Collectively the figure suggests that at Humboldt Bay wind shear is forcing the turbulence during both cloudy and clear-sky conditions, with HRRR model overestimating the winds and the wind shear. However, when the shear exponent is computed directly from the mean wind speeds (Table 2), HRRR slightly underestimates the average shear compared to the buoy data. This could be because that PDFs are dominated by high-wind events, which amplify apparent vertical gradients, while the mean shear exponent integrates over all wind-speed regimes and may therefore show weaker shear on average.

According to Table 1, mean 80-meter wind speed at Humboldt Bay is stronger than the one at Morro Bay regardless of weather conditions. The hub-height wind speed is stronger under clear-sky condition at both locations compared to that during cloudy conditions. We can relate clear-sky condition to stable stratification in atmospheric boundary layer. Under stable stratification, the surface layer—approximately 10% of the atmospheric boundary layer—becomes shallower in the marine boundary layer. This shallowness traps surface turbulent fluxes within the surface layer and decouples it from the layer above, leading to stronger wind speeds (Mahrt, 1999). These decoupled boundary layers are known to have lower cloudiness and higher winds than the coupled boundary layers (e.g. Serpetzoglou et al. 2008; Jones et al. 2011 etc.). The average wind speed difference between cloudy and clear-sky condition is stronger at Morro Bay, within both the buoy and HRRR records. During cloudy condition, HRRR is comparable to the observed hub-height wind speed. During clear sky condition, HRRR underestimates the hub-height wind speed at Morro Bay, while HRRR overpredicts it at Humboldt location.

Table 1 further demonstrates that the standard deviation of 80-meter wind speed during cloudy conditions is comparable between observations and HRRR at both locations. However, notable differences are observed under clear-sky conditions.

Table 1. Mean  $\pm$  standard deviation of wind speed of the lidar buoy observation and HRRR at 80-meter height for cloudy and clear conditions.

|        | Observed                       |                               | HRRR                           |                               |
|--------|--------------------------------|-------------------------------|--------------------------------|-------------------------------|
|        | Morro Bay [m s <sup>-1</sup> ] | Humboldt [m s <sup>-1</sup> ] | Morro Bay [m s <sup>-1</sup> ] | Humboldt [m s <sup>-1</sup> ] |
| Cloudy | 6.7±4.4                        | 8.5±4                         | 6.6±4.3                        | 8.5± 4                        |
| Clear  | 12±5.2                         | 9.5±3.4                       | 11.2±4.9                       | 11.1±4.9                      |

250






Following Wharton and Lundquist (2012), we compute the wind shear exponent  $\alpha$  following:

$$\alpha = \frac{\ln(v_2/v_1)}{\ln(z_2/z_1)}$$
 [1]

where  $v_2$  is the wind velocity at height  $z_2$  and  $v_1$  is the wind velocity at height  $z_1$ . For this analysis we have set  $z_2$ =160 m and  $z_1$  = 40 m. Table 2 lists  $\alpha$  values using the lidar buoy observations and HRRR for Morro Bay and Humboldt under cloudy, and clear conditions. As mentioned earlier,  $\alpha$  confirms that the shear is about 4 times stronger at Humboldt regardless of weather conditions. Stronger shear at Humboldt is also evident in the vertical profile of the lidar buoy wind speeds shown in Krishnamurthy et al. (2023). Under stable stratification, the surface layer is accompanied with weak turbulence, which lets stronger shear and veer to develop. Consistently, in both locations, shear is stronger under clear sky condition. The shear exponent is underestimated in HRRR at Humboldt location, while it is overestimated at Morro Bay. Previous modeling study by Zapata et al. (2021) showed that the cloud fraction decreases with an increase in near surface wind shear. Their results are consistent with our finding that cloud fraction in Humboldt (with stronger shear) is lower than in Morro Bay (Fig. 1).

Table 2. Mean  $\pm$  standard deviation of wind shear exponent  $\alpha$  of the lidar buoy observation and HRRR for cloudy and clear conditions.

|        | Observation |           | HRRR      |           |
|--------|-------------|-----------|-----------|-----------|
|        | Morro Bay   | Humboldt  | Morro Bay | Humboldt  |
| Cloudy | 0.05±0.10   | 0.2±0.22  | 0.07±0.13 | 0.15±0.14 |
| Clear  | 0.07±0.22   | 0.26±0.16 | 0.09±0.13 | 0.16±0.15 |

## 3.3 Characteristics of hub-height wind speed bias in terms of cloudiness







Fig. 5 shows the distribution of 80-meter wind speed bias in HRRR, binned by observed wind speed. Prior to binning, data was separated by its weather conditions (i.e., total, cloudy, and clear-sky). The median difference between the observed and modeled winds is less than 1 m s<sup>-1</sup> at both locations, while the range of the differences is higher at Morro Bay as compared to Humboldt Bay. At Morro Bay, the range of the bias across all observed wind speeds remains between  $\pm$  5 m s<sup>-1</sup>, while this range is higher at the Humboldt Bay. At Morro Bay, HRRR is largely unbiased for wind speeds below 10 m s<sup>-1</sup> but consistently underestimates the observed hub-height wind speed under cloudy conditions when wind speeds exceed this threshold, with the magnitude of underestimation increasing at higher wind speeds. During weak to moderate wind speed events (0 – 10 m s<sup>-1</sup>) the HRRR model slightly overestimates the wind speed under clear-sky conditions, and then the bias sign changes as the observed wind speed is beyond 10 m s<sup>-1</sup>. The change of the bias from positive to negative with increase in wind speed is the strongest under clear sky condition. The wind speed biases at Morro Bay during warm season are similar to those reported by Sheridan et al. (2022).

At Humboldt, HRRR slightly overestimates the hub-height wind speed when the observed wind speed is weak  $(0-5 \text{ m s}^{-1})$ , regardless of weather conditions. However, unlike Morro Bay, median HRRR bias for hub-height wind speed under clear-sky condition remains positive even during strong wind speed events, while it changes to negative under cloudy condition. Similar to Morro Bay, the bias under clear-sky and cloudy conditions increases with higher observed wind speed. Because of

the opposing bias signs between clear-sky and cloudy conditions at Humboldt, the median HRRR bias in hub-height wind speed over the entire period is close to zero. Our results are consistent with the findings in Sheridan et al. (2022), where the hub-height wind speed bias was investigated in terms of atmospheric stability. In Sheridan et al. (2022), the results by Rapid Refresh (RAP) indicated an overestimation of wind speed at Humboldt in stable conditions, while the bias flips its sign under unstable conditions. In their analysis, Morro Bay shows an underestimation of the wind speed regardless of the stability condition when RAP was compared to the lidar buoy observation. In both locations, the bias was much larger during the stable conditions, where the development of clouds is discouraged. However, we did not find any meaningful correlation between SST bias and wind-speed bias when examining the relationship over the entire period, as well as separately for cloudy conditions, clear-sky conditions, and stable atmospheric cases.

Figure 5 Box whisker plot of wind speed bias at 80 m between HRRR and Lidar (HRRR-Lidar) at various observed wind speed for clear-sky (red), all (grey) and cloudy (blue) conditions at Morro Bay (top) and Humboldt Bay (bottom).

We repeated the analysis shown in Fig. 5 for the entire period of available observation data (not shown). For Morro Bay, entire year data is utilized. For Humboldt, the period from December 2020 through May 2021 was excluded due to the unavailability of data. The analysis in the wind speed bias for longer period is consistent with that for warmer months.

Figure 6 Mean wind speed bias (HRRR - Lidar) for all weather condition in warm season (grey box), clear sky conditions (slanted stripes) and cloudy conditions (horizontal stripes) at various heights below 240m.


Next, we demonstrate how the mean wind speed bias changes with height (Fig. 6). At Morro Bay, HRRR underestimates the magnitude of low-level wind speed under all weather conditions, with the bias being most pronounced under clear-sky condition. The underestimation of wind speeds by the HRRR model increases with height with underestimation of -0.8 m s<sup>-1</sup> at 240 m. The HRRR model does simulate winds fairly accurately during cloudy conditions at the Morro Bay.

At Humboldt Bay, HRRR model during clear-sky condition overestimates the rotor layer wind speeds, and underestimates under cloudy condition. The overestimation of winds during clear-sky conditions decreases with height from 40 m to 240 m. Humboldt Bay experiences higher wind shear than Morro Bay. The HRRR model overestimates wind speeds during clear sky conditions at Humboldt Bay and underestimates them at Morro Bay. This suggests that the model has difficulty accurately simulating wind shear in clear sky situations, tending to overestimate winds when shear is strong and underestimate winds when shear is weak. Although this bias is much smaller during cloudy conditions, the HRRR model still overestimates wind speeds at Morro Bay and underestimates them at Humboldt Bay.

As discussed by Optis et al. (2016) and references therein, similarity-based wind-speed profile formulations derived from Monin-Obukhov similarity theory (MOST) can exhibit substantial bias under strongly stratified boundary layers. Because HRRR employs surface-layer and turbulence parameterizations that draw from this theoretical framework, such limitations may contribute to the under- or overestimation of wind speeds observed under stable, clear-sky conditions in our analysis. It should be noted, however, that HRRR applies MOST primarily within the surface layer, while turbulent mixing above this layer is represented using an eddy-diffusivity mass-flux (EDMF) approach that accounts for nonlocal transport and convective plumes (Olson et al., 2019a, 2019b). Optis et al. (2016) demonstrated that wind-speed bias at 40, 80, 140 and 200 m heights increased with the strength of stratification and even changed the sign under different stratification regime. Although informative, their results cannot be directly extrapolated to an offshore region with high cloudiness such as those studied here. Over land, boundary layer thermodynamic decoupling (stratification) is largely controlled by the presence or absence of surface fluxes as the turbulence is primarily modulated by these fluxes. In contrast, in offshore environments with cloud-topped boundary layer, the turbulence is primarily governed by radiative cooling at the cloud top. As a result, offshore boundary layers tend to become more decoupled during the day when cloud top cooling weakens and more coupled at night when radiative cooling is strongest. However, the degree of coupling can vary substantially depending on cloud optical thickness, synoptic forcing, and sea surface temperature gradients. In comparison, over land, boundary layers are typically coupled during the daytime when the surface heating is greatest and decoupled at night when the surface heating is minimal.

#### 3.4 Low level wind bias with marine stratocumulus clouds








Cases of warm stratocumulus clouds were identified using the cloud top temperature data from the GOES. The cases were identified from latitude-longitude maps of visible satellite imagery and cloud top temperatures. Three objective criteria, 1) extensive cloud cover for more than 2° (~200 km) in the zonal and meridional direction at the two locations, 2) cloud top temperatures in excess of 0°C, and 3) cloud cover lasting for more than 12 hours were enforced. Most of the cases had duration exceeding multiple days. Some cases included brief breaks in cloud cover lasting less than 3 hours and were therefore treated as part of the same event. Table 3 lists the duration of each identified marine stratocumulus cloud period along with its associated 80-meter wind bias, case-mean observed wind speed, and Pearson correlation coefficient. As previously discussed, hub-height wind speeds and the associated biases at Morro Bay are weaker than those at Humboldt. At both locations, HRRR generally underestimates the hub-height wind speed, except for the case in September at Morro Bay. During that event, the

mean wind speed is 4.1 m s<sup>-1</sup>. During weak wind  $(0 - 5 \text{ m s}^{-1})$  conditions, HRRR slightly overestimates the hub-height wind speed (Fig. 5). Additionally, the correlation between the time series of lidar buoy observations and HRRR wind speed is notably weaker at Humboldt (mean correlation coefficient r = 0.70) than at Morro Bay (r = 0.93), highlighting regional differences in model performance.

**Table 3.** List of stratocumulus events and their associated with observed mean wind speed at 80-meter, mean wind bias at 80-

meter, and correlation between HRRR and the lidar buoy.





| Site      | Duration (UTC)            | bias         | Correlation | Observed mean        |
|-----------|---------------------------|--------------|-------------|----------------------|
|           | YYYYMMDD:HH               | $[m s^{-1}]$ |             | wind speed           |
|           |                           |              |             | [m s <sup>-1</sup> ] |
| Morro Bay | 20210510:09 - 20210518:07 | -0.26        | 0.94        | 5.7                  |
|           | 20210619:02 - 20210709:00 | -0.04        | 0.97        | 7.8                  |
|           | 20210710:04 - 20210721:02 | -0.26        | 0.94        | 8.0                  |
|           | 20210802:14 - 20210816:21 | -0.27        | 0.95        | 7.7                  |
|           | 20210820:03 - 20210827:02 | -0.26        | 0.93        | 4.3                  |
|           | 20210828:15 - 20210908:07 | 0.03         | 0.89        | 4.6                  |
|           | 20210911:13 - 20210918:16 | -0.24        | 0.96        | 5.1                  |
|           | 20210922:00 - 20210927:21 | 0.24         | 0.89        | 4.1                  |
|           | Total hours: 2008         | -0.13        | 0.93        | 5.9                  |
| Humboldt  | 20210523:22 - 20210525:19 | 0.03         | 0.62        | 5.0                  |
|           | 20210620:20 - 20210707:18 | -0.64        | 0.86        | 7.4                  |
|           | 20210709:12 - 20210717:23 | -0.48        | 0.83        | 13.2                 |
|           | 20210724:11 - 20210726:18 | -0.57        | 0.45        | 12.8                 |
|           | 20210729:15 - 20210805:04 | -0.27        | 0.73        | 7.2                  |
|           | Total Hours: 1170         | -0.39        | 0.7         | 9.12                 |

Figure 7 presents the composite diurnal cycle of cloud top height (a-b), wind speed at 80 meter (c-d) and wind bias (e-f) at both Morro Bay and Humboldt Bay locations during stratocumulus periods listed in Table 3. At Morro Bay, GOES shows a deepening of the cloud layer in the morning while HRRR underestimates this diurnal change. At Humboldt Bay, GOES also indicates a slight deepening of clouds in the morning. These patterns are consistent with the warm season cloud characteristics shown in Fig.2. Wind speed, as shown in Fig 7 (c-d), exhibits a weak diurnal cycle at both locations. At Morro Bay, wind speeds peak in the evening whereas at Humboldt wind speeds increase at night. As indicated in Table 3, wind speeds are generally stronger at Humboldt Bay. The distribution of hub-height wind speed indicates that the bias at both locations is predominantly centered around zero. However, Humboldt exhibits a slightly greater underestimation of wind speed, which aligns with observations presented in Figures 5 and 6. For both Morro Bay and Humboldt, wind bias becomes positive during the nighttime. The range of the bias remains between ± 1.5 m s<sup>-1</sup> at Morro Bay while it exceeds this range at Humboldt (not shown). It was shown in Fig. 6 that stronger wind speed induces stronger bias.

Figure 7 Composite diurnal cycles of a-b) cloud top height, c-d) wind speed at 80-meter height, e-f) wind speed bias between lidar observation and HRRR during the stratocumulus periods listed in Table 3. Left column is from Morro Bay and right column is from Humboldt. Bar represents 25<sup>th</sup> and 75<sup>th</sup> percentile while line represents mean. Black lines are from GOES satellite data for cloud top height and lidar data for wind speed, while blue lines represent HRRR results. Green lines are directly computed as the difference between black and blues lines in panels (c) and (d).

Figure 8 a) timeseries of HRRR simulated cloud water mixing ratio (contour) and GOES reported cloud top height (black line), b) timeseries of wind speed at 80-meter observed by lidar buoy (black) and modeled by HRRR (red), and c) timeseries of wind speed bias (lidar – HRRR) at 80-meter. Location is Humboldt.



In Fig. 8, we closely examine the cloud top height and hub-height wind speed during one of the stratocumulus periods listed in Table 3. Fig. 8 (a) shows the timeseries of the vertical profile of the cloud water mixing ratio simulated by HRRR, alongside the estimated cloud top height by GOES. As shown in section 3.1, HRRR-simulated cloud tops are lower than the GOES-estimated values. While there is no clear indication of a diurnal cycle in cloud top height at Humboldt (Fig. 2b), during this particular period (2021/7/9 – 2021/7/17), GOES cloud top data shows a strong diurnal cycle for the first 5 days. In the observation, clouds are in the form of long-lived sheets while clouds simulated by HRRR is most of the time intermittent as indicated by the gaps in the cloud water mixing ratio. It is important to note that Figure 8 (a) shows only the resolved cloud liquid water mixing ratio fields from HRRR. The operational HRRR output does not include subgrid-scale cloud liquid water,

and therefore the figure cannot represent partially cloudy grid cells where unresolved cloud condensate may be present. As a result, some of the apparent "gaps" in the HRRR cloud field may in fact contain subgrid-scale cloud liquid that is not captured in the resolved output. We also compare the 80-meter wind speed between the lidar buoy observation and HRRR simulation. During this stratocumulus period, wind speed consistently exceeds the cut-in speed and occasionally surpassed the rated speed. The observation reveals a very strong diurnal cycle in hub-height wind speed and HRRR reproduces well in terms of amplitude and phase. However, both positive and negative bias exists in HRRR during this period, but the underestimation during the first 4 days outweighs some periods of overestimation (Fig. 8c). The diurnal cycle of the wind speed bias along with the diurnal cycle of the GOES reported cloud top height suggests the bias to be related to the cloud modulated turbulence in the boundary layer.

# 4. Summary and Conclusion








Marine stratocumulus clouds are shallow and located below the capping marine boundary layer inversion. The radiative cooling at cloud top often generates turbulence mixing downward, which can modulate the PBL turbulence profile (Wood, 2012). Also, the radiative cooling at the cloud top can strengthen the capping inversion, which can generate shear at the inversion layer. This shear is known to initiate atmospheric waves, which can adjust the wind speed and turbulence intensity within the marine boundary layer (Allaerts and Meyers, 2018). Therefore, better understanding the processes within the marine boundary layer bounded by marine stratocumulus clouds is important to better assess and predict wind conditions in coastal regions dominated by these clouds (Shaw et al., 2022).

There are wind farm lease areas off the coast of Northern and Central California (Musial et al., 2019), where marine stratocumulus clouds are abundant during warmer months (Wood, 2012). In the fall of 2020, two DOE buoys with multiple instruments including Doppler lidar and pyranometer were deployed for about a year to these lease areas (Humboldt and Morro Bay locations) to continuously measure the vertical profile of wind speed below 240 meters and global broadband solar radiation, among others. The presence of this dataset, in combination of GOES satellite date, provides a unique opportunity to assess the HRRR model bias in forecasting the wind speed in rotor-layer relevant heights in terms of cloudiness. The primary findings from this work are as follows.

- a) Consistent with previous studies, GOES satellite data indicate that marine stratocumulus clouds are prevalent at both locations during the warmer months. Cloud cover is greater at Morro Bay compared to Humboldt Bay, and Morro Bay also exhibits a more pronounced diurnal cycle in cloud top height. Overall, the HRRR model captures the general cloudiness well but underestimates both the cloud-top heights and the amplitude of the diurnal cycle at both sites, with the underestimation being especially pronounced at Morro Bay.
- b) Wind speeds and shear within the rotor layer are stronger at Humboldt Bay than at Morro Bay. At both locations, rotor layer wind speeds and shear are higher on clear-sky days compared to cloudy days. The HRRR model generally reproduces these patterns, although some biases remain.

- c) At both locations the HRRR model tends to underestimate the 80-meter wind speed in cloudy condition (with a median bias of -1.5 m s<sup>-1</sup>), and this negative bias increases with higher wind speeds. Under clear-sky conditions, The wind speed bias is larger with the model overestimating wind speeds at Morro Bay (positive bias) and underestimating them at Humboldt Bay (negative bias).
- d) Under cloudy conditions, the mean wind speed bias at different heights (40, 80, 160, and 240 meters) is generally small, close to zero, or shows slight underestimation by the model. Under clear-sky conditions, the opposite bias pattern between Humboldt Bay and Morro Bay remains consistent at all rotor-layer heights.
  - e) Stratocumulus cloud events at Morro Bay and Humboldt Bay during warmer months were identified by visually inspecting GOES images. Those specific stratocumulus events also reveal that mean wind speed and wind speed bias are stronger at Humboldt Bay. In both locations, the mean bias is negative.

Due to the shallow depth of marine boundary layer, the top of the surface layer, where Monin-Obukhov (M-O) similarity theory is applicable, could be close to the wind turbine rotor layer. Also, M-O similarity theory is strongly dependent upon the atmospheric stability. Many studies have shown that M-O similarity theory does not work very well above surface layer and under stable stratification. This may explain why the HRRR bias is stronger under clear-sky conditions. The different HRRR bias sign at the two locations under clear-sky condition may be related to the strength of atmospheric stability. Further measurement such as boundary layer thermodynamic profile would be required to verify this theory.

Our analysis uses data only for May through September time-period because that is when marine stratocumulus clouds are most abundant in the region. When we include entire data period including colder months, we get a very similar results as shown in Fig. 6, even though the data period from December through May is missing for Humboldt location: 1) bias is larger under clear-sky conditions than in cloudy condition, 2) bias sign under clear-sky condition is opposite between Humboldt and Morro Bay. Although the fraction of clear-sky days is much lower compared to the cloudiness fraction during warmer months, in colder months, the fraction of clear-sky days increases (not shown). Therefore, it is equally important to improve HRRR model physics to better represent marine boundary layer processes under clear-sky (stable) condition.

Lastly, in this article we characterized the biases in the HRRR model simulated rotor layer winds, and clouds. Broadly the analysis shows the wind speed biases to differ between cloudy and clear-sky conditions, and significant bias in the HRRR simulated cloud fraction and cloud top heights. The turbulence in the cloud-topped boundary layers is modulated by cloud processes, which then in-turn affects the boundary layer thermodynamic stability and winds. To what degree the inaccuracies in representation of clouds affects the representation of winds and vice-versa can only be understood through model simulations. This will be the focus of our work in the near future.

#### Data availability






Buoy data used in this study are available at <a href="https://a2e.energy.gov">https://a2e.energy.gov</a>. All data can be provided by the author upon request.

## 440 Author Contribution

All authors contributed to the development of the analysis concept and reviewed the manuscript. JL performed the analysis and wrote the manuscript. AM compiled and provided the GOES and CERES data. LM compiled and provided the buoy data.

#### **Competing Interests**

The authors declare that they have no competing interests.

### Acknowledgements

Funding for this study was provided by the U.S. Department of Energy (DOE) Office of Energy Efficiency and Renewable Energy, Wind Energy Technologies Office. The views expressed in this article do not necessarily represent those of the DOE or the U.S. Government. VG and AM were supported by the DOE Wind Energy Technologies Office (WETO), Office of Energy Efficiency and Renewable Energy (EERE), under Contract DE-AC02-06CH11357 awarded to Argonne National Laboratory. This work was performed under the auspices of the U.S. Department of Energy by Lawrence Livermore National Laboratory under Contract DE-AC52-07NA27344. LLNL-JRNL-2006801. The language of this manuscript was improved with the assistance of LLNL's LivChat AI service.




450

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
