# Peer review of "Characterization of HRRR simulated Rotor Layer Wind Speeds and Clouds along Coast of California"

_Wind Energy Science, 2025_

## Author Comment (AC1)

This paper uses roughly 1-year of surface-based doppler lidar wind observations collected at two locations off the west coast of California (Morro Bay and Humboldt Bay) to evaluate the 3-hourly wind speed forecasts from the High-Resolution Rapid Refresh (HRRR) numerical weather prediction model at several different heights from 40 to 240 m above the sea surface. They have demonstrated that overall, the HRRR's performance matches the observations pretty well, with better agreement at Morro Bay vs Humboldt Bay, but that there are conditions where the model has significant wind speed errors.

Generally speaking, this paper reads very well. The figures are clean, the writing is clear, the results are largely well supported. I have listed a number of comments / suggestions below that I'd like the authors to consider and address before the paper could be accepted for publication.

We would like to sincerely thank you as a reviewer for your thoughtful and constructive comments. Your insights have greatly helped us improve the clarity and overall quality of the manuscript. We have carefully addressed all suggestions in blue and believe the revised version benefits substantially from your valuable feedback.

**Major comments:**

Line 102: what is the accuracy and uncertainty of the GOES derived cloud top heights? Is there any seasonal dependency to these values?

GOES measures cloud-top temperature (CTT) with an accuracy of approximately 1 K, and CTH is then inferred from lapse-rate calculations following the method of Sun-Mack et al. (2014). The bias in CTT retrievals was assessed by Yu et al. (2012), who showed that while CTTs are generally accurate, uncertainties in the lapse rate often translate into larger errors in derived CTHs. Several approaches have attempted to mitigate this issue. Zuidema et al. (2009) used sea-surface temperature (SST)-based constraints to improve CTH estimation, but this approach also showed mixed performance. A comparison study by Ghate et al. (2019) demonstrated that all major CTH retrieval techniques perform well under certain conditions and fail under others, with no clear gold standard. The most recent SATCORPS Edition 4 GOES algorithm represents the best-performing version to date; Yost et al. (2021) found mean GOES-CloudSat/CALIOP CTH differences of about 100 m.

In our study, the mean HRRR-GOES CTH differences (~150–200 m) are somewhat larger than the expected GOES uncertainty, suggesting that the discrepancy likely reflects HRRR deficiencies rather than retrieval error. Because stratocumulus clouds off the California coast are typically thin (< 200 m) and the boundary-layer lapse rate closely

follows the dry adiabat, a 1 K uncertainty in retrieved CTT corresponds to roughly 100 m in CTH uncertainty—consistent with Yost et al. (2021).

Finally, as noted in Mitra et al. (2025), a seven-year comparison between GOES and ERA5 cloudy boundary-layer heights indicated ERA5–GOES CTH differences of  $-0.17 \pm 0.62$  km (Humboldt) and  $-0.22 \pm 0.51$  km (Morro Bay). Together, these findings suggest that while GOES retrievals have inherent uncertainties, their magnitude is small compared with the systematic HRRR biases discussed in this study. As shown in Mitra et al. (2021), a seasonal dependence in GOES CTH uncertainty is possible, but its effect is minor relative to these overall differences.

We added the following paragraph in Section 2.2 to provide readers information about uncertainty and seasonality related to the GOES cloud retrievals:

GOES determines cloud-top height (CTH) from infrared-retrieved cloud-top temperature (CTT) and lapse-rate calculations following Sun-Mack et al. (2014). The CTT measurements have an accuracy of about 1 K (Yu et al., 2012), which corresponds to an uncertainty of roughly 100 m in CTH for a dry-adiabatic lapse rate. Previous studies have shown that inaccuracies in lapse-rate estimates can lead to CTH errors of this magnitude (Zuidema et al., 2009; Ghate et al., 2019). Comparisons of the most recent SATCORPS Edition 4 GOES retrievals with CloudSat and CALIOP indicate mean differences near 100 m (Yost et al., 2021). A recent seven-year analysis comparing ERA5 and GOES data reported ERA5–GOES differences of  $-0.17 \pm 0.62$  km (Humboldt) and  $-0.22 \pm 0.51$  km (Morro Bay) (Mitra et al., 2025). These results imply that GOES retrieval uncertainties are on the order of 100–200 m—smaller than the HRRR–GOES differences reported here—and that the HRRR bias in CTH likely reflects model rather than retrieval limitations.

Sun-Mack, S., Minnis, P., Chen, Y., Kato, S., Yi, Y., Gibson, S.C., Heck, P.W. and Winker, D.M. (2014) Regional apparent boundary layer lapse rates determined from CALIPSO and MODIS data for cloud-height determination. Journal of Applied Meteorology and Climatology, 54, 990–1011. <a href="https://doi.org/10.1175/JAMC-D-13.081.1">https://doi.org/10.1175/JAMC-D-13.081.1</a>.

Yu, F., Wu, X., Raja, M.K.R.V., Li, Y., Wang, L. and Goldberg, M.D. (2013) Diurnal and scan angle variations in the calibration of GOES imager infrared channels. IEEE Transactions on Geoscience and Remote Sensing, 51, 671–683

Zuidema, P., Painemal, D., DeZoeke, S.P. and Fairall, C.W. (2009) Stratocumulus cloudtop height estimates and their climatic implications. Journal of Climate, 22, 4652–4666.

C. R. Yost, P. Minnis, S. Sun-Mack, Y. Chen and W. L. Smith, "CERES MODIS Cloud Product Retrievals for Edition 4—Part II: Comparisons to CloudSat and CALIPSO," in

IEEE Transactions on Geoscience and Remote Sensing, vol. 59, no. 5, pp. 3695-3724, May 2021, doi: 10.1109/TGRS.2020.3015155.

Mitra, A., Ghate, V., Krishnamurthy, R.: Wind and Climate Variability within the Californian Offshore Wind Energy Areas, Argonne Technical Report, ANL/NSE-24/26, Pending Release at U.S. Department of Energy (DOE) OSTI, https://doi.org/10.2172/2568064, 2025

Mitra, A., Di Girolamo, L., Hong, Y., Zhan, Y., & Mueller, K. J. (2021). Assessment and error analysis of Terra-MODIS and MISR cloud-top heights through comparison with ISS-CATS lidar. Journal of Geophysical Research: Atmospheres, 126(9), e2020JD034281. <a href="https://doi.org/10.1029/2020JD034281">https://doi.org/10.1029/2020JD034281</a>

Ghate, V. P., Mechem, D. B., Cadeddu, M. P., Eloranta, E. W., Jensen, M. P., Nordeen, M. L., and Smith, W. L.: Estimates of entrainment in closed cellular marine stratocumulus clouds from the MAGIC field campaign, Q. J. Roy. Meteor. Soc., 145, 1589–1602, https://doi.org/10.1002/qj.3514, 2019

Line 102: how are multi-layer cloud systems handled? Certainly there are cases where there is an upper level cirrus cloud that obscures the lower atmosphere?

In cases of multilayer cloud systems (e.g., thin cirrus overlying thicker low clouds), the GOES infrared retrieval typically reports a cloud-top height (CTH) corresponding to the upper cirrus layer. Such retrievals yield CTHs higher than our low-cloud thresholds and are therefore automatically excluded from our analysis as false "mid-level" or "high-cloud" detections. We have confirmed that these filtering criteria effectively remove most multilayer cases from the dataset.

We acknowledge that in some situations, cirrus clouds may be optically thick enough to completely obscure the lower cloud layer, preventing detection by GOES infrared channels. However, these high-cloud retrieval periods are not used to draw conclusions about the marine boundary layer or to evaluate HRRR wind-speed biases, as our analysis focuses solely on periods classified as low-cloud or clear-sky conditions.

To incorporate this information, in the revised manuscript section 2.2, we added the following paragraph:

Instances of multilayer cloud systems (e.g., thin cirrus overlying low stratocumulus) were handled by filtering out retrievals with cloud-top heights exceeding our low-cloud thresholds. In such cases, the GOES infrared retrieval reports the upper-level cirrus as the primary cloud top, leading to spuriously high CTH values that are automatically excluded from the analysis. Situations with optically thick cirrus that fully obscure the

low-level cloud deck are likewise classified as high-cloud conditions and are not used when evaluating boundary-layer properties or HRRR wind-speed biases.

Line 111 regarding your "minor discrepancies" comment: could this be a definition and/or sensitivity issue? For example, the GOES can only identify a cloud layer if the optical depth is above some threshold (which might depend on other atmospheric conditions), whereas the model might define a cloud if there are any hydrometeors in a volume. How was this handled? Was an instrument simulator applied to the HRRR's output to try to mimic the GOES observations? (I suspect not, but you should at least be clear that this is an issue that could impact these analyses).

Thank you for raising this question. The minor discrepancies between the GOES- and HRRR-derived cloud-top heights and cloud fractions may indeed partly reflect differences in how clouds are defined and detected in each system. No satellite observation simulator (e.g., an instrument forward operator) was applied to the HRRR output in this study. Therefore, some portion of the observed differences between HRRR and GOES results likely arises from these definitional and sensitivity differences. We have revised the manuscript to explicitly acknowledge this as a potential source of uncertainty in the cloud comparisons. The added sentences are shown below:

These differences may partly reflect the distinct definitions of cloud used in the two datasets. The GOES retrieval identifies a cloud layer only when the optical depth exceeds a threshold that varies with viewing geometry and atmospheric conditions, while HRRR defines clouds based on the simulated presence of hydrometeors within model grid cells. Because no satellite instrument simulator was applied to the HRRR output, such definition and sensitivity differences might contribute to the observed discrepancies in cloud-top height and cloud fraction.

Line 134: One of the limitations of this study is that the liquid water path (LWP) cannot be compared between the GOES and model easily. My sense is that the model has too little LWP in its cloud, and thus in the daytime there is too little absorption of shortwave radiation, and thus too little diabatic heating in the cloud, which is why the cloud did not deepen. Your suggestion that it could be an issue with longwave radiative cooling is also possible, but again, this connects to errors in getting the diurnal evolution of the LWP in the cloud correct. So please expand this discussion a bit more.

Thank you for raising this question. We agree that one limitation of our study is the lack of direct comparison of liquid water path (LWP) between GOES and HRRR, which constrains our ability to diagnose the model's representation of cloud radiative processes. The absence of a diurnal cycle in HRRR cloud-top height may indeed result from insufficient simulated LWP. A low LWP would reduce daytime shortwave

absorption within the cloud, leading to too little diabatic heating and limiting cloud deepening, while also affecting longwave cooling at night. Therefore, both underestimation of daytime heating and misrepresentation of nocturnal longwave radiative cooling could contribute to the weak diurnal variability of simulated cloud-top height. We have expanded the discussion in Section 3.1 to reflect this broader interpretation and to acknowledge that errors in HRRR's representation of the diurnal LWP cycle likely underlie its muted diurnal cloud-top height variability. The added sentences in the revised manuscript are shown below:

One likely contributing factor is the model's underrepresentation of cloud liquid water path (LWP), which cannot be directly compared with GOES in this study. A low simulated LWP would reduce the amount of shortwave absorption within the cloud during the daytime, resulting in insufficient diabatic heating and limited cloud deepening. Conversely, too little condensate would also weaken longwave cooling at night, reducing turbulent mixing at cloud top. These combined effects could explain the muted diurnal variation in HRRR cloud-top height and indicate that biases in the model's treatment of cloud microphysics and radiative processes jointly contribute to the discrepancy.

Line 230: You are hypothesizing, here and later in the paper, that the model is not capturing stability properly and that could be the source of the bias. However, error in the sea surface temperature (SST), which is a boundary condition for the model, is another possibility. And it is possible that there are different errors in SST in Morro vs Humboldt Bays, especially in clear skies? That would affect the stability profile, which would then feed back into the turbulence and wind profiles. I believe the buoys have SST observations on them, and it would be a pretty straight-forward analysis to determine if SST biases are correlated with the wind biases. This might result in your need to update the statement at line 350.

Thank you for the suggestion. We conducted additional analyses to examine whether SST biases are correlated with wind-speed biases. We evaluated scatter plots for all time periods, for cloudy conditions, for clear-sky conditions, and for cases with stable stratification (defined when the buoy-measured air-sea temperature difference, Tair – SST, is positive). In all cases, we found no clear correlation between SST bias and wind-speed bias, as shown below:

In the revised manuscript, we added this information as given below:

However, we did not find any meaningful correlation between SST bias and wind-speed bias when examining the relationship over the entire period, as well as separately for cloudy conditions, clear-sky conditions, and stable atmospheric cases.

Line 253: you state "similarity-based wind speed profile model". The way this is worded suggests that the HRRR is using this approach. Similarly, in line 348, you give the same impression. The HRRR uses Monin-Obukhov theory in the surface layer; however, above the surface layer the HRRR uses an eddy diffusivity mass flux approach. See papers by Olson et al (BAMS 2019) and NOAA Tech Memo (https://doi.org/10.25923/n9wm-be49).

Thank you for your comment. We revised the sentences to avoid the confusion and is given as below in the revised manuscript:

As discussed by Optis et al. (2016) and references therein, similarity-based wind-speed profile formulations derived from Monin–Obukhov similarity theory (MOST) can exhibit substantial bias under strongly stratified boundary layers. Because HRRR employs surface-layer and turbulence parameterizations that draw from this theoretical framework, such limitations may contribute to the under- or overestimation of wind speeds observed under stable, clear-sky conditions in our analysis. It should be noted, however, that HRRR applies MOST primarily within the surface layer, while turbulent mixing above this layer is represented using an eddy-diffusivity mass-flux (EDMF) approach that accounts for nonlocal transport and convective plumes (Olson et al., 2019a,2019b).

Figure 8 and the discussion starting at line 299: There are both resolved clouds (i.e., where the entire model grid cell is cloudy) and subgrid-scale clouds (i.e., where there is partial cloudiness in a grid cell). Unfortunately, the HRRR does not save the liquid water content operationally to an output file. Thus, this figure can only show the resolved cloud liquid clouds. It is possible that there is significant subgrid-scale cloud liquid between the "gaps" shown in Fig 8a and discussed at line 305, but we just don't know for sure. You can get the LWP from the HRRR from the HRRR data archive on the AWS cloud server; the LWP is the vertical integral of both the resolved and subgrid-scale liquid water content profiles.

Thank you for bringing up this point. We attempted to extract the liquid water path (LWP) from the HRRR output available to us but were unable to locate this variable in our dataset. If you could provide additional guidance on how to retrieve LWP from the HRRR archive, we would be glad to explore this further for the revision.

In the meantime, we have added the following sentences to the manuscript to clarify the limitations associated with using the cloud water mixing ratio variable:

It is important to note that Figure 8 shows only the resolved cloud liquid water mixing ratio fields from HRRR. The operational HRRR output does not include subgrid-scale cloud liquid water, and therefore the figure cannot represent partially cloudy grid cells where unresolved cloud condensate may be present. As a result, some of the apparent "gaps" in the HRRR cloud field may in fact contain subgrid-scale cloud liquid that is not captured in the resolved output.

**Minor comments:**

Line 13: "multiple instruments"

Thank you. I corrected it

Line 14: there are many types of lidar. Please say "including Doppler lidar"

Thank you. I followed your suggestion

Line 54: ...resource assessments over the United States. (the HRRR's domain)

Thank you. I added "over the United States"

Line 80: Isn't GOES-17 the only satellite that is relevant for this paper?

Yes, since we are only using the time frame of the DOE buoys. That line was in reference to the entire SATCORPS dataset, not the subset of the data used in question. To clarify the confusion, we added a following sentence in the revised manuscript:

For this study, GOES-17 is the only satellite providing data that overlaps with the buoy observation period.

Line 130: "in both locations" --> "in either location"

Thank you. I corrected it.

Line 131: I believe you have this reversed, as the GOES shows higher values in the daytime in Fig 2

Thank you for raising the question. The sentence was meant for the typical marine stratocumulus situation, which we do not see in our GOES analysis. To avoid the confusion, I revised the sentence as shown below:

Under typical marine stratocumulus conditions, cloud tops are higher at night and lower during the day due to stronger nocturnal cloud-top radiative cooling, our GOES analysis (Fig. 2) shows the opposite behavior at both sites; the reason for this discrepancy is presently unclear.

Figure 3: please use the same y-axis range for all panels within the figure, as it would make it much easier to compare the different panels

Thank you for the suggestion. We now have the same y-axis panels for all panels in the revised manuscript. We also changed the color scheme as it previously didn't pass the color blindness test. Here is the remade figure 3.

Figure 4: same comment

Thank you for the suggestion. We now have the same y-axis panels for all panels in the revised manuscript. We also changed the color scheme as it previously didn't pass the color blindness test. Here is the remade figure 4.

Line 173: "overestimates wind shear" – this is not supported by Table 2, which shows the opposite

Thank you for raising this question. We acknowledge the discrepancy in the behavior of wind shear, which is qualitatively evident in Fig. 4 and quantitatively summarized in Table 2. We believe this difference arises from the way the data were processed to derive the PDFs and the shear exponent. For the shear exponent, we used time-averaged wind speeds, whereas the PDFs include the full range of instantaneous wind speeds within each weather regime. To avoid confusion, we have added the following clarification to the manuscript:

However, when the shear exponent is computed directly from the mean wind speeds (Table 2), HRRR slightly underestimates the average shear compared to the buoy data. This could be because that PDFs are dominated by high-wind events, which amplify apparent vertical gradients, while the mean shear exponent integrates over all wind-speed regimes and may therefore show weaker shear on average.

Line 214: "model consistently underestimates" – I disagree. Fig 5 top panel shows that the HRRR is essentially unbiased for wind speeds less than 10 m/s at Morro Bay

Thank you for raising this question. We revised the questioned sentence in the revised manuscript as given below:

At Morro Bay, HRRR is largely unbiased for wind speeds below 10 m s-1 but consistently underestimates the observed hub-height wind speed under cloudy conditions when wind speeds exceed this threshold, with the magnitude of underestimation increasing at higher wind speeds.

Figure 6: why didn't you use the same style of plot as in Figure 5? It would make the paper more consistent and easier to read

Figure 5 presents box-and-whisker plots of the wind-speed bias at 80 m under all, clear-sky, and cloudy conditions, while Figure 6 shows the mean wind-speed bias at multiple heights (40, 80, 160, and 240 m). We wanted to show how the mean wind bias changes with height in Figure 6.

Line 261: "decoupled during the day" – is this always true (i.e., for every cloudy day in your analysis)? I would be surprised if this was true

We agree that the statement about offshore boundary layers being "decoupled during the day" should not be interpreted as a universal condition. Our intention was to describe the general tendency in marine stratocumulus-topped boundary layers, where radiative cooling at the cloud top weakens during daytime, leading to partial decoupling between the surface and cloud layers. However, the degree of decoupling depends on local thermodynamic and dynamical conditions and may vary among cases. We have revised the text to clarify this point in the revised manuscript and the change is given below:

As a result, offshore boundary layers tend to become more decoupled during the day when cloud top cooling weakens and more coupled at night when radiative cooling is strongest. However, the degree of coupling can vary substantially depending on cloud optical thickness, synoptic forcing, and sea surface temperature gradients.

Line 275: the differences in the correlations between the two locations are important. Modify the end of this sentence to indicate the mean values of the correlation coefficient to help strengthen this point

Thank you for your suggestion. I modified the sentence in the revised manuscript and is given below:

Additionally, the correlation between the time series of lidar buoy observations and HRRR wind speed is notably weaker at Humboldt (mean correlation coefficient r = 0.70) than at Morro Bay (r = 0.93), highlighting regional differences in model performance.

Figure 7: the blue minus black lines in panes c and d do NOT equal the results shown in panels e and f. Please recompute and update the figure

Thank you for raising this question. We believe that the blue minus black lines in panels (c) and (d) differ from the results shown in panels (e) and (f) because we first composited the bias for each hour across the analysis period and then computed the composite mean for each hour. This approach may explain why the difference between the composite means is not identical to the mean of the composite differences. To avoid the confusion, we replotted (e) and (f) to have blue minus black lines in panel c and d. The new figure is in the revised manuscript and also given below.

---

## Author Comment (AC2)

**Synopsis:**

The manuscript with the title "Characterization of HRRR simulated Rotor Layer Wind Speeds and Clouds along Coast of California", which was submitted for publication to the journal Wind Energy Science by the authors Jungmin Leem, Virendra P. Ghate, Arka Mitra, Lee M. Miller, Raghavendra Krishnamurthy and Ulrike Egerer, deals with the evaluation of wind speed and cloud forecasts by the HRRR model for two offshore sites off the coast of California. Data from 3-hour forecast from HRRR are compared with data collected with lidars and additional meteorological sensors on two buoys as well as with satellite data. According to the results presented by the authors the observed seasonal cycle of cloud top height is well reproduced at both sites by the HRRR model. However, in the warm season stratocumulus cloud top heights are underestimated by HRRR, especially at one of the two sites. Another finding by the authors is that clear sky conditions come along with larger wind speeds at the the two sites investigated than cloudy conditions. Clear-sky conditions come also along with a larger bias of the wind speeds predicted by HRRR, although the sign of the bias is different at the two sites for which the authors did the analysis.

**Evaluation:**

Understanding how the presence of clouds inside the atmospheric boundary layer changes the wind speed in the atmospheric boundary layer is a topic of high relevance for the wind energy community. The understanding of such processes is a key for modeling these processes correctly and thus enabling also an improved modeling of wind conditions. Thus, in my opinion the topic of the manuscript is interesting for the wind energy community and is basically suitable for being published in Wind Energy Science. However, I have a few comments on the current version of the manuscript that might support sharpening of the clarity of the paper. Moreover, I have a couple of minor comments. Therefore, my recommendation is that in the next step the authors should apply some corrections to the current version of the manuscript before a decision on its publication in Wind Energy Science can be taken.

We would like to sincerely thank you as a reviewer for your thoughtful and constructive comments. Your insights have greatly helped us improve the clarity and overall quality of the manuscript. We have carefully addressed all suggestions in blue and believe the revised version benefits substantially from your valuable feedback.

**Comments on the content:**

**1: Abstract: From my point of view the authors could and should present the objectives of the paper more clearly in the abstract.**

Following your suggestion, I revised the abstract section. The revised abstract is provided below in my response to comment #2.

**2: Abstract: The authors claim that "the findings from this study will potentially inform how to improve the modeling of wind resources off the coast of Northern California". However, there are no clear advices for the next steps towards improving the modeling of wind resources given in the manuscript. Therefore, I think that either the respective sentence in the abstract should be revised or the conclusions part of the manuscript should be extended with corresponding content.**

Following your advice, I revised the abstract to have a clear objective that is not for the model improvement. Below is the revised abstract.

Stratocumulus clouds, with their low cloud base and top, affects the atmospheric boundary layer wind and turbulence profile, thereby modulating wind energy resources. GOES satellite data reveal an abundance of stratocumulus clouds in late spring and summer months off the coast of Northern and Central California, where there are active plans to deploy floating offshore wind farms at two lease areas (near Morro Bay and Humboldt). From fall 2020, two buoys equipped with multiple instruments including Doppler lidar were deployed for about one year in these wind farm lease areas to assess the rotor-layer wind conditions in these locations. The objective of this study is to evaluate how well the High Resolution Rapid Refresh (HRRR) model represents stratocumulus cloud characteristics and turbine-relevant rotor-layer winds (surface to 300 m) by comparing HRRR simulations with buoy and satellite observations. We first find that the HRRR model reproduces the seasonal cycle of cloud-top height reasonably well in these regions. However, during the warm season especially at Morro Bay—the HRRR-simulated stratocumulus clouds tend to have lower tops by about 150 m and exhibit weaker diurnal cycles than satellite observations. Our analysis also shows that rotor-layer wind speeds and vertical shear are stronger at Humboldt than at Morro Bay, and both are generally stronger under clear-sky conditions. Finally, the HRRR model bias in rotor-layer wind speed is small under cloudy conditions but larger and dependent on observed wind speed under clear skies. Specifically, HRRR underestimates wind speeds at Morro Bay and overestimates them at Humboldt under clear-sky conditions.

**3: General comment: The literature review presented in the introduction is rather short. A more extensive introduction e.g. into stratocumulus-topped ABL could be valuable for the reader. I think e.g. from the paper by Kopec et al. (2016) some relevant information on stratocumulus-topped ABLs could be presented. Concerning the HRRR model I'm missing some examples of previous studies that evaluated that model (especially for wind energy purposes, but also information on other evaluation studies**

might be interesting). Moreover, the model itself could be presented in more detail. E.g., what are the initial and boundary conditions used?

Kopec, M. K., Malinowski, S. P., Piotrowski, Z. P., 2016: Effects of wind shear and radiative cooling on the stratocumulus-topped boundary layer, Q. J. R. Meteorol. Soc., 142, 3222-3233, https://doi.org/10.1002/qj.2903

Thank you so much for the suggestion. I added findings from Kopec et al (2016) in the introduction to provide detailed processes in stratocumulus-topped ABLs. Below is the paragraph added in the Introduction:

Recent modeling and observational studies have provided deeper insight into the physical processes governing the stratocumulus-topped boundary layer (STBL). In particular, Kopec et al. (2016) investigated how radiative cooling at the cloud top and wind shear across the capping inversion jointly influence turbulence generation and cloud-top structure using large-eddy simulations based on the POST field campaign. Their analysis demonstrated that radiative cooling intensifies convective circulations within the STBL, while wind shear enhances turbulence and mixing in the inversion layer above the cloud, often producing a distinct turbulent sublayer that is dynamically decoupled from the convective motions below. These findings highlight the importance of representing both shear- and radiation-driven turbulence for accurate modeling of stratocumulus dynamics and their influence on boundary-layer winds. Such processes are particularly relevant for offshore wind resource assessment, where low-level clouds modulate turbulence and wind shear within the turbine rotor layer.

I also added a paragraph in the revised introduction section with a couple examples of HRRR evaluation in the wind energy context. The added paragraph is provided below:

Several prior studies have evaluated the HRRR model relevant to hub-height wind speed assessment. Liu et al. (2025) benchmarked hub-height wind speeds from HRRR analyses against multi-source observations across the southeastern United States and found that wind-speed biases were strongly influenced by local topography and land surface characteristics such as forest canopy height. Complementary efforts under the Second Wind Forecast Improvement Project (WFIP2) assessed experimental updates to the HRRR model using scanning Doppler lidar measurements in the complex terrain of the Columbia River Basin (Pichugina et al., 2020). That study demonstrated that the HRRR model's wind-speed errors were largest below about 150 m above ground level and that improvements in model physics and grid resolution led to modest but consistent reductions in mean wind-speed bias. Together, these findings highlight the importance of evaluating HRRR performance in diverse environments, including

offshore regions where boundary-layer cloud processes can further influence rotorlayer winds.

Also, HRRR description is extended, and the change is presented in the response of minor comment #8.

Liu, Y., et al. (2025). Benchmarking near-surface winds in the HRRR analyses using multi-source observations over complex terrain. Journal of Applied Meteorology and Climatology. <a href="https://doi.org/10.1175/JAMC-D-24-0163.1">https://doi.org/10.1175/JAMC-D-24-0163.1</a>

Pichugina, Y., Banta, R., Brewer, W., Bianco, L., Draxl, C., Kenyon, J., Lundquist, J., Olson, J., Turner, D., Wharton, S., Wilczak, J., Baidar, S., Berg, L., Fernando, H. J. S., McCarty, B., Rai, R., Roberts, B., Sharp, J., Shaw, W., ... Worsnop, R. (2020). Evaluating the WFIP2 Updates to the HRRR Model Using Scanning Doppler Lidar Measurements in the Complex Terrain of the Columbia River Basin. Journal of Renewable and Sustainable Energy, 12(4), Article 043301. https://doi.org/10.1063/5.0009138, https://doi.org/10.1063/5.0009138

**4: Line 53/54: "... despite HRRR being one of the most widely used forecasting tools in wind energy resource assessments" Please add references for this statement. Is this forecasting tool applied for wind resource assessment (i.e. derivation of wind speed information for periods with a length of decades)? My understanding is that mostly models run in hindcast mode are used for wind resource assessment studies. The limited time available for producing a forecast limits e.g. the time that can be spent on the data assimilation process. Moreover, later on it is stated that HRRR is applied for wind power forecasting. I would see this as something different from a wind resource assessment. HRRR does not output wind power directly. Thus, there might be wind power forecasting systems that make use of HRRR results for generating a wind power forecast, but I think using HRRR alone will not allow you to do a wind power forecast. To summarize, I ask the authors to add clarifications concerning the mentioned points to their manuscript.**

You have the valid point. We apologize for any confusion with the wording "wind resource assessment". We clarified it by replacing "wind resource assessment" with something in line of "skill in representing rotor-layer wind conditions" throughout the manuscript.

The reference for "... despite HRRR being one of the most widely used forecasting tools in wind energy resource assessments" is provided as part of the response to comment #3.

**5: Line 94/95: "The HRRR reported wind profiles were linearly interpolated ..." Why has a linear interpolation been applied. In the surface layer the wind profile is expected to be logarithmic. Wouldn't it therefore make more sense to apply a logarithmic interpolation in this area? Another possibility would be to apply a power law profile for the wind speed and use this for interpolation.**

You are right about this. However, the heights where HRRR data is available (for instance 38.4, 87.5, 166.9 and 242.4 meter height) was close to the HRRR model level. Also, in the rotor-layer range, the logarithmic curvature is small especially over ocean or weakly stratified boundary layers. Therefore, we thought there would be a minimal impact from the interpolation method.

**6: Line 97-99: The current description "we extract the corresponding lidar data for each hour" could still be made more precise, e.g. as follows: As HRRR provides data only at a full hour we also use only 10-minute averaged lidar data with the same timestamp as that of the forecast.**

Thank you for the suggestion. We revised the sentence following your suggestion as shown below:

Because HRRR outputs are available only at full-hour intervals, we selected the 10-minute averaged lidar measurements whose timestamps correspond exactly to each forecast hour, ensuring consistent temporal alignment between the HRRR time series and the buoy observations.

**7: How do the authors deal with the fact of HRRR and lidar data being effectively different types of averaged data? E.g. does 10 minute averaged lidar data compare better to HRRR than 30 minute averaged lidar data? Did the authors check for a possible phase shift between the lidar and the model data? Why did the authors decide not to interpolate the model data in the horizontal directions of space to the position of the lidar measurements?**

HRRR hourly output is an instantaneous snapshot, not a temporal average. Accordingly, we expect 10-minute-averaged LiDAR data to compare better with HRRR than 30-minute averages. HRRR's horizontal resolution is 3 km, and the mean separation between the LiDAR buoy and the HRRR extraction point is ~2.5 km at Morro Bay and ~1.4 km at Humboldt. Although we could average over neighboring HRRR grid cells, the already fine 3-km resolution suggests such spatial averaging would have little impact on our results.

**8: Figure 3: The following comment is also connected to my impression that the manuscript could benefit from improving the clarity of its objectives. Is HRRR evaluated**

with respect to its potential for resource assessment or wind power forecasting? Is the PDF of the wind speed the best quantity to assess the performance of a tool used for forecasting the wind speed? Even if the PDFs of the model and the measurements looked perfectly the same, this would not necessarily mean that it Is well suited for the purpose of wind power forecasting. For that purpose other parameters like the absolute bias would be more relevant. Thus, if the objective were evaluating HRRR for the purpose of wind power forecasting I would recommend to not starting the evaluation with presenting the PDFs.

Following your suggestions in Comments #1, #2, and #4, we have clarified the objective of this manuscript to focus on evaluating the HRRR model for its skill in representing rotor-layer wind conditions. We have also replaced the term "wind resource assessment" with "rotor-layer wind speed assessment" throughout the text. Therefore, we would like to retain the PDFs in their current form, as they provide a useful overview of the rotor-layer wind characteristics.

**9: Line 172: "This suggests that HRRR model overestimates wind shear at Humboldt Bay location compared to the observations" The authors derive this statement from the PDFs of the wind speeds at different heights. I'm wondering why the authors do not directly investigate and present the wind shear itself before they conclude on it. I think this would strengthen the statements made on the wind shear.**

Thank you for this valuable comment. Table 2 in the manuscript summarizes the shear exponent metric used to quantify the wind shear strength directly. From this analysis, we confirm that the shear is stronger at Humboldt and under clear-sky conditions. However, the shear exponent does not indicate that HRRR overestimates the shear at Humboldt, as was inferred from the PDFs in Figure 4. We believe this apparent difference arises from the distinct sampling approaches—PDFs emphasize higher wind-speed regimes, while the shear exponent is based on time-averaged wind speeds. We have updated the text in Section 3.2 to explicitly reference Table 2 and clarify that the discussion of wind-shear bias is based on the direct shear-exponent calculations rather than solely on the PDF comparisons. The revised sentence is provided below:

In Fig. 4, we repeated the analysis shown in Fig. 3 for Humboldt Bay location. At Humboldt, the lidar buoy measurements of wind speed in the rotor layer shows a more spread-out distribution spanning 0 – 20 m s-1, while HRRR reported winds range from 0 to 30 m s-1. The HRRR model simulates a peak in the PDF (Figure 4a) that shifts toward higher speeds with increasing heights, unlike what is observed (Figure 4d). At Humboldt, strong vertical shear is present regardless of cloud conditions, as the increase in wind speed with height is evident in both datasets, but HRRR model consistently overestimates this increase. During clear-sky conditions, the wind speed

tends to be stronger compared to cloudy conditions, with peaks around 10 – 18 m s-1. Collectively the figure suggests that at Humboldt Bay wind shear is forcing the turbulence during both cloudy and clear-sky conditions, with HRRR model overestimating the winds and the wind shear. However, when the shear exponent is computed directly from the mean wind speeds (Table 2), HRRR slightly underestimates the average shear compared to the buoy data. This could be because that PDFs are dominated by high-wind events, which amplify apparent vertical gradients, while the mean shear exponent integrates over all wind-speed regimes and may therefore show weaker shear on average.

**10: Line 253-254: "As stated in Optis et al. (2016) and the references therein, similarity-based wind speed profile models produce a large bias under strongly stratified boundary layer." Is it the aim here to provide an explanation for the bias in strongly stratified boundary layers observed by the authors? In that case the link between HRRR and similarity theory should be made clearer.**

Thank you for this helpful comment. Our intention was indeed to suggest a possible explanation for the HRRR wind-speed bias under strongly stratified boundary layers. HRRR, like most numerical weather prediction models, uses parameterizations for surface-layer turbulence that are based on Monin–Obukhov similarity theory (MOST). As discussed in Optis et al. (2016), such similarity-based formulations can become inaccurate under stable or strongly stratified conditions, where turbulence is weak and the assumptions of constant flux and stationarity break down. We have revised the text to explicitly state this link between HRRR's parameterization framework and the potential source of bias and is shown below:

As discussed by Optis et al. (2016) and references therein, similarity-based wind-speed profile formulations derived from Monin–Obukhov similarity theory can produce substantial bias under strongly stratified boundary layers. Because the HRRR model employs surface-layer and turbulence parameterizations grounded in this framework, such limitations likely contribute to the under- or overestimation of wind speeds observed under stable, clear-sky conditions in our analysis.

**11: Line 269: "Some cases had breaks in the cloud for less than 3 hours that got filled" I think the authors should elaborate on this a little bit further. For me the meaning of this statement was unclear. What was filled with what?**

Thank you for your comment. We revised the sentence as provided below:

Some cases included brief breaks in cloud cover lasting less than 3 hours and were therefore treated as part of the same event.

**12: Figure 8b): Is there an explanation why the lidar data, although it is averaged over 10 minutes, is fluctuating quite strongly (and seemingly with some preferred frequency), while the model data is comparatively smooth?**

Thank you for raising this question. We discovered that the jumps in the lidar data were caused by a timestamp error, which has been corrected in the revised manuscript and also is given below. The observed time series are not expected to appear perfectly smooth, as atmospheric turbulence naturally induces wind-speed fluctuations that the model does not fully capture. Therefore, it is reasonable that the HRRR results appear smoother than the observations, even at the 10-minute time scale.

**13: Line 317-319: "Therefore, better understanding the processes within the marine boundary layer bounded by marine stratocumulus clouds is important to better assess and predict wind resources for offshore wind farms in California." Is it necessary to limit this statement to offshore wind farms off the coast of California?**

Thank you for your comment. We revised the sentence as given below:

Therefore, better understanding the processes within the marine boundary layer bounded by marine stratocumulus clouds is important to better assess and predict wind conditions in coastal regions dominated by these clouds

**Minor comments:**

**1: Line 20: Please change "... those fields are stronger ..." to " ... those parameters have larger values ...".**

**Changed following your suggestion**

**2: Line 40: Please change "... were made from a research flights" to " ... were made from research flights".**

**Changed following your suggestion**

**3: Line 46: Please change "Staring October 2020" to "Starting in October 2020".**

**Changed following your suggestion**

**4: Line 59: Please change "model wind bias in relation to of various" to "model wind bias in relation to".**

**Changed following your suggestion**

**5: Line 62: Please delete "at the Humboldt location".**

**Changed following your suggestion**

**6: Line 71: Please change "from from 40" to "from 40".**

**Changed following your suggestion**

**7: Line 75: I think the sentence "Temporal cloud mask is estimated to be cloudy" should be rephrased. What is a temporal cloud mask? This term needs to be better introduced.**

**Following your suggestion, I modified the sentence as shown below:**

The temporal cloud mask, which provides a timeseries of sky conditions, was derived by quantifying the deviation between the observed and modelled clear-sky collar radiation. When the measured radiation was more than 10% lower than the modeled

clear-sky value, the condition was classified as cloudy. Otherwise, it was designated as cloud-free (Krishnamurthy et al., 2023).

**8: Section 2.3: The description of HRRR should be extended. In the current text there is no information on the data assimilation process that is used in the model. It would be good to add references to previous evaluation studies that support that HRRR provide forecast with high-fidelity as stated in the description by the authors.**

Thank you for your suggestion. We extended HRRR description (section 2.3) by including initial and boundary data for the model as well as the data assimilation method. We also provide references to previous HRRR evaluation studies such as Liu et al. 2005; Pichugina et al. 2020 to the text and the change is provided in the response to major comment #3. Below is the addition in section 2.3.

HRRR's initial conditions are generated from the parent Rapid Refresh (RAP) model one hour prior to forecast initialization. RAP forecast is advanced forward, assimilating the most recent observations to provide a dynamically consistent state for HRRR initialization. Lateral boundary conditions for HRRR are supplied by RAP model at three-hour intervals. HRRR employs a hybrid ensemble-variational (EnVar) data-assimilation system that integrates a 36-member ensemble to represent flow-dependent background errors. The assimilation cycle updates every hour and incorporates a wide range of conventional, satellite, and radar observation. Within each cycle, radar reflectivity and radial-velocity data are assimilated every 15 minutes to improve the representation of cloud and convective structures. Surface and soil states are updated using short-term forecasts from the HRRR Data Assimilation System (HRRRDAS), which maintains temporal continuity between analysis cycles. This framework is designed to minimize spin-up errors and improve the depiction of boundary-layer and mesoscale processes critical to wind-forecast applications.

**9: Line 92-94: The corresponding sentence should be revised to avoid a possible misunderstanding. "... are close to the heights of the lidar measurement, which is at 40, 80, 160 and 240 meters" should therefore be changed to "... are close to the heights of 40, 80. 160 and 240 m at which data from lidar measurements is available"**

**Changed following your suggestion**

**10: Line 116-118: "In December 2020, a large wave event at Humboldt buoy location results in the power outage and lead to a large data gap. The machines on the buoy were back online on May 25th, 2021, at Humboldt location." This information had been provided in the manuscript before. Therefore, there is a reduncancy that should be removed.**

**Removed those sentences following your suggestion.**

**11: Line 138, line 142: References are made to a figure S1, however, there is no figure S1 in the manuscript. Please correct these references.**

Figure S1 refers to the figure in supplement material, which can be found here: https://wes.copernicus.org/preprints/wes-2025-108/wes-2025-108-supplement.pdf

**12: Figure 3: The following comment is also connected to my impression that the manuscript could benefit from improving the clarity of its objectives. Is HRRR evaluated with respect to its potential for resource assessment or wind power forecasting? Is the PDF of the wind speed the best quantity to assess the performance of a tool used for forecasting the wind speed? Even if the PDFs of the model and the measurements looked perfectly the same, this would not necessarily mean that it Is well suited for the wind power forecast. Other parameters like e.g. the absolute bias would be more relevant. Thus, if the objective were evaluating HRRR for the purpose of wind power forecasting I would suggest not starting with the presentation of the PDFs of wind speed.**

Thank you for your suggestion. Following your earlier comments, we have revised and clarified the objective of the study to focus on evaluating the HRRR model for its representation of rotor-layer wind characteristics in the area of interest. We have decided to retain the PDFs in their current position, as they effectively introduce the discussion of wind characteristics.

**13: Line 260: Please change "As a results ..." to " As a result".**

**Changed following your suggestion**

**14: Line 343: Please change "... wind bias ..." to "wind speed bias".**

Changed following your suggestion

---

## Referee Report (RR1)

I would like to thank the authors for thoroughly revising their manuscript and providing detailed answers to all the comments that were made by the two reviewers. I do not see the need for further modifications to the manuscript. Therefore, I can recommend to accept the manuscript as it is for publication.